# On Parameter Stability Region of LADRC for Time-Delay Analysis with a Coupled Tank Application

**Dazi Li *** **, Xun Chen, Jianqing Zhang and Qibing Jin**

Institute of Automation, Beijing University of Chemical Technology, Beijing 100029, China; xunchen@mail.buct.edu.cn (X.C.); zhangjq@mail.buct.edu.cn (J.Z.); jinqb@mail.buct.edu.cn (Q.J.)

**\*** Correspondence: lidz@mail.buct.edu.cn; Tel.: +86-10-6443-4930

**Abstract:** The control of time-delay systems is a hot research topic. Ever since the theory of linear active disturbance rejection control (LADRC) was put forward, considerable progress has been made. LADRC shows a good control effect on the control of time-delay systems. The problem about the parameter stability region of LADRC controllers has been seldom discussed, which is very important for practical application. In this study, the dual-locus diagram method, which is used to solve the upper limit of the LADRC controller bandwidth, is studied for both first-order time-delay systems and second-order time-delay systems. The characteristic equation roots distribution is firstly transformed into the problem of finding the frequency of the dual-locus diagram intersection point. To solve the problem for second-order time-delay system LADRC controllers, which is a dual 10-order nonlinear equation, a transformation has been made through Euler's formula and genetic algorithm (GA) has been adopted to search for the optimal parameters. Simulation results and experimental results on coupled tanks show the effectivity of the proposed method.

**Keywords:** LADRC; dual-locus diagram; GA; PLC; parameter stability region

---

## 1. Introduction

In most industrial processes such as level control, boiler temperature control, and internal pressure control of distillation columns, the time-delay phenomenon is widespread. Because of the limitations of measuring devices, energy conversion devices, and other reasons, time delay is inevitable in most industrial processes. The mixing process of viscous liquid is a typical industrial process with pure time delay [1]. The change in pump speed cannot immediately generate liquid viscosity at the outlet of pipeline change because of the delay in transmission of the viscosity signal and liquid mixing and transportation. Another example is temperature control in beer fermentation, which is a common process with an extremely long delay time [2]. To sum up, when the setting value changes, the controlled variable cannot be tracked and stabilized in the setting value in time. This phenomenon leads to asynchrony between input and output. When the controlled plant is in the closed loop of the process with such a time delay, its own dynamic characteristics are affected, and consequently, the system vibrates easily or even tends to diverge. Such controlled objects are disadvantageous to the design of controllers. Therefore, how to deal with a time-delay system is a major problem in the control field.

A modern method is to estimate and compensate the time delay in real time [3]. At present, most of the time-delay problems in industry are solved with the proportional-integral-differential (PID) algorithm, although numerous modern control algorithms are available. In addition to the difficulties in implementing these algorithms, their stability is an important factor. PID controllers can

be designed without specific plant models, and the PID algorithm has strong adaptability because of its robustness. The algorithm is also easy to implement, and therefore is widely used in industry. In the 1990s, Han proposed the idea of active disturbance rejection control (ADRC) based on the advantages of the PID algorithm [4]. Through an in-depth analysis of the development of control theory, Han examined the nature of control, that is, to detect the error between the setting and actual values, and to eliminate the error with certain methods [5]. According to this idea, Han discussed the problems in the design of control law which depended on a model based on his understanding of model theory and cybernetics [6]. At the same time, by analyzing how the control algorithm can reject the disturbance in completely unknown circumstances, Han proposed a new concept of estimating compensation for the error, which subsequently evolved into ADRC [7].

Since ADRC was proposed, abundant achievements have been reported in previous decades. On the side of stability analysis, ADRC is also improving consistently. Zhao and Guo [8] designed the ADRC algorithm for single-input single-output (SISO) systems and analyzed its convergence. For the stability of the closed-loop system, Chen, Sun and Yang [9] proposed the stability theory of linear active disturbance rejection control (LADRC). Tian and Gao [10] provided the expression of the transfer function, which also promoted the development of ADRC in the field of stability analysis. Robustness analysis of ADRC in the frequency domain was also conducted. With the proposed expression of LADRC transfer function, Wu and Chen [11] presented the transfer function expression of nonlinear ADRC. The two expressions provide additional convenience in the development of ADRC theory. By summarizing the previous studies, Huang and Xue [12] provided the design of ADRC and identified the design differences for various control problems. Madonski [13] proposed general error-based ADRC for swift industrial implementations. Furthermore, these studies provided valuable experience in the application process, and increased the convenience in the application of ADRC. ADRC technology has been used in motors, reactors, heaters and other industrial plants [14–16].

For time-delay systems, Li, Ai and Tian [17] proposed method of ADRC combined with feedforward control to compensate the delay time. Fu and Tan [18] use the method of ADRC to control unstable processes with time delays. Stability analysis is an important step if ADRC is applied to the actual process. Progress has been made on the qualitative analysis method of ADRC in stability analysis, and the method for determining the parameter stability region of the first-order time-delay system LADRC controller has also been proposed [19]. Sufficient simulation verification has been carried out [20], but it has not been verified in the actual process. The parameter stability region of LADRC controllers in higher-order time-delay systems is not discussed.

In this study, the dual-locus diagram method for solving the parameter stability region of the first-order time-delay system LADRC controllers is validated on the actual device by programmable logic controller (PLC). The closed-loop control loop constructed by the second-order time-delay system and second-order LADRC is analyzed. The process of obtaining the parameter stability region of the LADRC for the second-order time-delay systems is introduced. The dual-locus diagram method can provide the range of the parameter. In this range, genetic algorithm (GA) is proposed to search for the optimal parameter. Combined with the dual-locus diagram method, the theory of solving the stability region of the controller parameters is given. Finally, it is verified in the simulation.

## 2. Linear Active Disturbance Rejection Control (LADRC)

To facilitate the subsequent theoretical derivation and application of this study, LADRC is selected as the main control algorithm. The LADRC structure is shown in Figure 1.

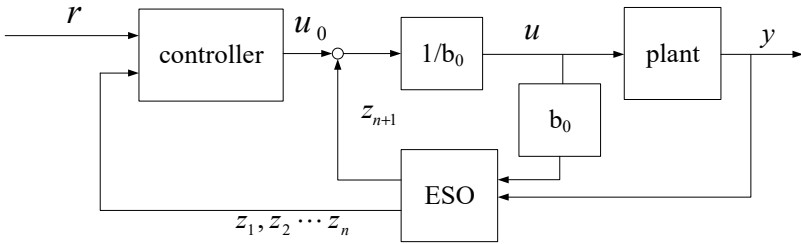

**Figure 1.** Block diagram of LADRC.

We consider the following SISO system:

$$y^{(n)} = bu + f\left(y, \dot{y}, \cdots y^{(n-1)}, \omega, t\right) \tag{1}$$

where $u$, $y$, $\dot{y}$, $y^{(n-1)}$, $b$, and $\omega$ are the input and output of the system, the derivative of the output of the system, the $n-1$ order derivative of the output of the system, the system gain, and the disturbance, respectively. These variables are transformed into the following "integral" form:

$$y^{(n)} = (b - b_0)u + f + b_0 u = \overline{f} + b_0 u \tag{2}$$

where $\overline{f} = (b - b_0)u + f$ indicates total disturbance, $b_0$ is a rough approximation of $b$, and the function of LADRC is to estimate and compensate $\overline{f}$. We define state variable $X = \begin{bmatrix} x_1 & x_2 & \cdots & x_n & x_{n+1} \end{bmatrix}^T$, where $x_{n+1}$ is the extended state.

Transforming the preceding equation into the form of state space, we can obtain

$$\begin{cases} \dot{x}_1 = x_2 \\ \quad \vdots \\ \dot{x}_{n-1} = x_n \\ \dot{x}_n = x_{n+1} \\ \dot{x}_{n+1} = \dot{\overline{f}} = h \\ \quad y = x_1 \end{cases} \tag{3}$$

The above equation is written in the following matrix form:

$$\begin{cases} \dot{X} = AX + b_0 Bu + Eh \\ \quad y = CX \end{cases} \tag{4}$$

where $A = \begin{bmatrix} 0 & 1 & 0 & \cdots & 0 \\ 0 & 0 & 1 & \cdots & 0 \\ \vdots & \vdots & \vdots & \ddots & \vdots \\ 0 & 0 & 0 & \cdots & 1 \\ 0 & 0 & 0 & \cdots & 0 \end{bmatrix}$, $B = \begin{bmatrix} 0 \\ 0 \\ \vdots \\ 1 \\ 0 \end{bmatrix}$, $C = \begin{bmatrix} 1 \\ 0 \\ \vdots \\ 0 \\ 0 \end{bmatrix}$, $E = \begin{bmatrix} 0 \\ 0 \\ \vdots \\ 0 \\ 1 \end{bmatrix}$.

The total disturbance $\overline{f}$ is accurately estimated and compensated by the extended state observer (ESO), which can be designed in the following form:

$$\begin{cases} \dot{Z} = AZ + b_0 Bu + L(y - \hat{y}) \\ \quad \hat{y} = CZ \end{cases} \tag{5}$$

where $b_0$ is also an adjustable parameter when the system gain of the controlled plant is unknown. $L = [l_1 \ l_2 \ \cdots \ l_n \ l_{n+1}]^T$ is the gain matrix of ESO, $Z = [z_1 \ z_2 \ \cdots \ z_n \ z_{n+1}]^T$ is the output of ESO, and $z_{n+1}$ is the estimate of $\overline{f}$. We find out from the overall design process that the design of LADRC is basically

the same. First, the states contained in the system are defined, and then all the disturbances of the system, which are different from the standard type, are observed by extending a single state, thereby realizing the compensation for the disturbance.

After obtaining the ESO, we select the control rate as follows:

$$u = \frac{u_0 - z_{n+1}}{b_0} \tag{6}$$

For the output of the controller $u_0$, we choose:

$$u_0 = k_1(r - z_1) - k_2 z_2 - \cdots - k_n z_n \tag{7}$$

where $k_1, k_2, \cdots k_n$ are the undetermined coefficients and $r$ is the reference signal.

## 3. Parameter Stability Region Determination of LADRC

In this section, we discuss the principle of the dual-locus diagram method and parameter stability region determination method of LADRC on the general mathematical model of linear time-delay systems. The parameter stability region determination method is discussed by using the example of second-order time-delay systems.

### 3.1. Principle of Dual-Locus Diagram Method

In process control engineering, many theoretical analysis methods have been proposed for single-variable control in a single loop. For such systems, the main method for stability analysis is to solve their closed-loop characteristic equation

$$1 + Q(s) = 0 \tag{8}$$

where $Q(s)$ is the open-loop transfer function and $s$ is the Laplace operator.

In many modern control systems, the closed-loop characteristic equations are generally not described by Equation (8), and this equation is not generally representative. If multiloop and multivariable situations occur in the control loop, such as chemical process control, UAV flight control, and so on, this expression will not describe the plant. To solve this problem, we can take the following general expression:

$$L_1(s) + L_2(s) = 0 \tag{9}$$

where $L_1(s)$ and $L_2(s)$ are functions related to the Laplace operator $s$ and satisfying Equation (9), and may contain a nonlinear part. Making the transposition for Equation (9), we can have the following equation:

$$L_1(s) = -L_2(s) \tag{10}$$

In this manner, the left and right sides of the equation are depicted in the $s$ plane, and the two Nyquist curves can be obtained. From the preceding process, we can see that Equations (9) and (10) are equivalent; thus, the system characteristics described by the two equations have not changed. Therefore, the system stability can be analyzed by performing stability discrimination on the trajectories at both ends of Equation (10).

In the time-delay system, the open-loop transfer function $Q(s)$ can be written as $G_c(s)G_0(s)e^{-\tau s}$, where $G_c(s)$ is the transfer function of the controller and $G_0(s)$ is the transfer function of the plant without time delay $e^{-\tau s}$. Then the closed-loop characteristic equation $1 + Q(s) = 0$ can be written as $G_c(s)G_0(s) = -e^{\tau s}$, where $G_c(s)G_o(s)$ and $e^{\tau s}$ correspond to $L_1(s)$ and $L_2(s)$, respectively. The stability of the original system can be determined by analyzing the interaction between the two trajectories. The main theory used here is the dual-locus diagram method. In the next derivation, we use $L(s)$

instead of $L_1(s)$ to facilitate the derivation process. We quote the following theorem of stability criterion in a time-delay system with the dual-locus diagram method:

**Lemma 1. [21]** *If the system is stable, then the closed-loop characteristic equation of the system must satisfy one of the following conditions:*

*If the Nyquist curve of $L(s)$ is non-intersecting with the Nyquist curve of $-e^{\tau s}$, then $L(s)$ has no poles in the right half of the s plane.*

*If the Nyquist curve of $L(s)$ intersects with the Nyquist curve of $-e^{\tau s}$, then $L(s)$ has no poles in the right half of the s plane, and the Nyquist curve of $L(s)$ arrives at the point of intersection earlier than $-e^{\tau s}$, which means that $\varphi(L(j\omega_i)) > \varphi(-e^{j\omega_i \tau})$.*

### 3.2. Parameter Stability Region Determination Method of LADRC

The dual-locus diagram method is mainly used in the frequency domain, and the transfer function of the control algorithm is needed. Only when this transfer function is known can the closed-loop transfer function of the entire control system be solved. Thus, the stability region of the controller parameters can be obtained by using the dual-locus diagram method. The next step is to derive the transfer function of LADRC. As the controlled plant is mainly a second-order time-delay system, we select the second-order LADRC. The following figure shows the transfer function structure of LADRC.

In this figure, $R(s)$ is the reference signal, $Y(s)$ is the output signal, $D(s)$ is the uncertain external disturbance signal, $U(s)$ is the control signal, $G_p(s)$ is the controlled plant, and $H(s)$ and $G_c(s)$ are the undetermined terms of the controller. We can obtain the control signal from Figure 2 as follows:

$$U(s) = (R(s)H(s) - Y(s))G_c(s) = R(s)H(s)G_c(s) - Y(s)G_c(s) \tag{11}$$

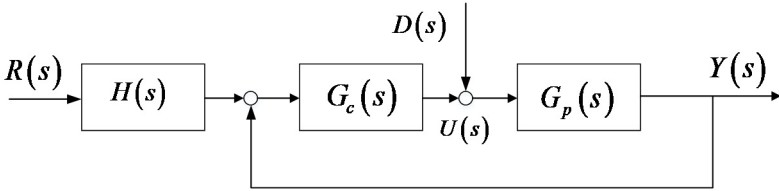

**Figure 2.** Transfer function block diagram of LADRC.

The control law corresponding to the second-order system is selected as follows:

$$U(s) = \frac{1}{b_0}(k_1(R(s) - Y(s)) - k_2 Z_2(s) - Z_3(s)) \tag{12}$$

where $k_1$ and $k_2$ are the undetermined coefficients. According to Equation (5), we can obtain

$$\dot{Z} = (A - LC)Z + b_0 Bu + Ly \tag{13}$$

Laplace transformation of the above equation is

$$
\begin{aligned}
Z(s) \quad &= (sI - A + LC)^{-1} \begin{bmatrix} b_0 B & L \end{bmatrix} \begin{bmatrix} U(s) \\ Y(s) \end{bmatrix} \\
&= \frac{1}{s^3 + l_1 s^2 + l_2 s + l_3} \begin{bmatrix} b_0 s & l_1 s^2 + l_2 s + l_3 \\ b_0(s^2 + l_1 s) & l_2 s^2 + l_3 s \\ -l_3 b_0 & l_3 s^2 \end{bmatrix} \begin{bmatrix} U(s) \\ Y(s) \end{bmatrix}
\end{aligned} \tag{14}
$$

where $I$ is the unit diagonal matrix and $l_1$, $l_2$, $l_3$ are undetermined coefficients. From the preceding discussion, we can obtain:

$$\begin{cases} Z_2(s) = \frac{b_0\left(s^2 + l_1 s\right)U(s) + \left(l_2 s^2 + l_3 s\right)Y(s)}{s^3 + l_1 s^2 + l_2 s + l_3} \\ Z_3(s) = \frac{-l_3 b_0 U(s) + l_3 s^2 Y(s)}{s^3 + l_1 s^2 + l_2 s + l_3} \end{cases} \tag{15}$$

Substituting Equation (14) into Equation (12), we have

$$U(s) = \frac{1}{b_0}\left[ k_1\left(R(s) - Y(s)\right) - k_2 \frac{b_0\left(s^2 + l_1 s\right)U(s) + \left(l_2 s^2 + l_3 s\right)Y(s)}{s^3 + l_1 s^2 + l_2 s + l_3} - \frac{-l_3 b_0 U(s) + l_3 s^2 Y(s)}{s^3 + l_1 s^2 + l_2 s + l_3} \right] \tag{16}$$

Comparing the equation above and Equation (11), we find that both are expressions of $U(s)$ and their coefficients must be equal, so $H(s)$ and $G_c(s)$ must satisfy the following conditions:

$$\begin{cases} H(s) = k_1 \times \frac{s^3 + l_1 s^2 + l_2 s + l_3}{k_1 s^3 + (k_1 l_1 + l_2 k_2 + l_3)s^2 + (k_1 l_2 + k_2 l_3)s + k_1 l_3} \\ G_c(s) = \frac{1}{b_0} \times \frac{k_1 s^3 + (k_1 l_1 + l_2 k_2 + l_3)s^2 + (k_1 l_2 + k_2 l_3)s + k_1 l_3}{s^3 + (l_1 + k_2)s^2 + (l_2 + k_2 l_1)s} \end{cases} \tag{17}$$

where "$\times$" is the multiplication operator.

According to the theory of bandwidth parameterization [22], the correspondence between bandwidths and unknown variables in the preceding equation is as follows:

$$\begin{cases} l_1 = 3\omega_o \\ l_2 = 3\omega_o^2 \\ l_3 = \omega_o^3 \\ k_1 = \omega_c^2 \\ k_2 = 2\omega_c \end{cases} \tag{18}$$

where $\omega_o$ is the observer bandwidth and $\omega_c$ is the controller bandwidth. Thus, $H(s)$ and $G_c(s)$ can be written as the expressions of $\omega_o$ and $\omega_c$ in the following form:

$$\begin{cases} H(s) = \omega_c^2 \times \frac{s^3 + 3\omega_o s^2 + 3\omega_o^2 s + \omega_o^3}{\omega_c^2 s^3 + \left(3\omega_c^2 \omega_o + 6\omega_c \omega_o^2 + \omega_o^3\right)s^2 + \left(3\omega_c^2 \omega_o^2 + 2\omega_c \omega_o^3\right)s + \omega_c^2 \omega_o^3} \\ G_c(s) = \frac{1}{b_0} \times \frac{\omega_c^2 s^3 + \left(3\omega_c^2 \omega_o + 6\omega_c \omega_o^2 + \omega_o^3\right)s^2 + \left(3\omega_c^2 \omega_o^2 + 2\omega_c \omega_o^3\right)s + \omega_c^2 \omega_o^3}{s^3 + (3\omega_o + 2\omega_c)s^2 + \left(3\omega_o^2 + 6\omega_c \omega_o\right)s} \end{cases} \tag{19}$$

In Equation (19), $b_0$ is another unknown variable. If the second-order LADRC controller estimates the states of the controlled system accurately, and the unknown variable $b_0$ can be approximately equal to the steady-state gain $b$ of the controlled plant, then only two parameters are in Equation (18). In the theory of bandwidth parameterization, the observer bandwidth $\omega_o$ is usually 3–5 times larger than the controller bandwidth $\omega_c$ in the process of actual tuning parameters, which means that $\omega_o = 3 \sim 5\omega_c$. In this way, only one parameter is left to be tuned. Through the detailed introduction, the concrete expression of the second-order LADRC transfer function has been derived. The next main object is the controlled plant, which is the transfer function of the second-order time-delay system. Thus, a complete control system is constructed, and then the stability region of the controller parameters is obtained by using the dual-locus diagram method. A second-order time-delay system $T_1 T_2 \frac{d^2 y(t)}{dt} + (T_1 + T_2)\frac{dy(t)}{dt} + y(t) = bu(t - \tau)$ is considered, and its transfer function in frequency domain is as follows:

$$G_p(s) = \frac{b}{(T_1 s + 1)(T_2 s + 1)}e^{-\tau s} \tag{20}$$

where $b$ is the system gain, $T_1$ and $T_2$ are the time constants, and $\tau$ is the delay time. For the second-order time-delay system described, we select the second-order LADRC as the control algorithm. Equation (18)

provides the specific parameters. If accurate $b$ is estimated, then $b_0$ is equal to $b$, and we can obtain the following closed-loop transfer function of the system:

$$G_{cl}(s) = \frac{H(s)G_c(s)G_p(s)}{1 + G_c(s)G_p(s)} \tag{21}$$

From the above transfer function, we can find the following closed-loop characteristic equation:

$$\delta = 1 + G_c(s)G_p(s) = 1 + G_c(s)G_0(s)e^{-\tau s} \tag{22}$$

From the empirical formulas mentioned, we can see that the observer bandwidth is generally 3–5 times the controller bandwidth. Here, we select $K$, that is, $\omega_o = K\omega_c$, so that we can analyze the stability region later.

Let $L(s) = G_c(s)G_0(s)$, and substitute Equations (18) and (19) into Equation (21), and then the expression can be derived as follows:

$$L(s) = \frac{\omega_c^2 s^3 + \left(3K\omega_c^3 + 6K^2\omega_c^3 + K^3\omega_c^3\right)s^2 + \left(3K^2\omega_c^4 + 2K^3\omega_c^4\right)s + K^3\omega_c^5}{s\left(s^2 + 3K\omega_c s + 2\omega_c s + 3K^2\omega_c^2 + 6K\omega_c^2\right)(T_1 s + 1)(T_2 s + 1)} = -e^{\tau s} \tag{23}$$

**Theorem 1.** *For the second-order time-delay system $\frac{b}{(T_1 s+1)(T_2 s+1)}e^{-\tau s}$, the second-order LADRC is selected as the control algorithm to build a complete closed-loop control system, which is shown in Figure 2. $L(s)$ is also obtained according to Equation (23).*

*We consider the following conditions:*

(a)   *The steady-state gain $b$, time constants $T_1$ and $T_2$, and delay time $\tau$ are given explicitly.*
(b)   *The appropriate bandwidth ratio K in the designed LADRC is given explicitly.*

*Under the preceding conditions, we can draw the following conclusions according to Nyquist stability criterion [23]:*

(1)   *If the open-loop transfer function of the system has no poles in the right half plane and the Nyquist curve of $L(j\omega)$ and $-e^{j\omega\tau}$ do not intersect, the system is stable.*
(2)   *If a positive real root satisfies $L(j\omega) = 1$ and the phase angle of $L(j\omega)$ is larger than that of $-e^{j\omega\tau}$ at their intersection frequency, the closed-loop system is stable and the stability region of the controller bandwidth $\omega_c$ can be calculated accurately as follows:*

$$\phi = \left\{\omega_c \middle| \varphi_1 - \varphi_2 > 0, \tau, T_1, T_2, b, K\right\} \tag{24}$$

*where $\varphi_1$ and $\varphi_2$ are the phase angles of $L(j\omega)$ and $-e^{j\omega\tau}$, respectively.*

**Proof.** To obtain the intersection frequency $\omega_i$, the following equation can be solved:

$$\left|L(j\omega_i)\right| = \left|-e^{j\omega_i\tau}\right| = 1 \tag{25}$$

From Equation (23), we can know that the order of $L(s)$ is 5; thus, the order of $\left|L(j\omega_i)\right|$ is 10. Equation (25) can be simplified as follows:

$$a\omega_i^{10} + b\omega_i^8 + c\omega_i^6 + d\omega_i^4 + e\omega_i^2 - 1 = 0 \tag{26}$$

where $a,b,c,d$, and $e$ are the functions of $\omega_c$. Naturally, $\omega_i$ can be regarded as a function of $\omega_c$, $K$, $T_1$, and $T_2$, that is, $\omega_i = f(\omega_c, K, T_1, T_2)$. Substituting $\omega_i$ into $L(j\omega_i)$ and $-e^{j\omega_i\tau}$, we can obtain their phase angles at the intersection frequency $\omega_i$, where $\varphi_2 = \tau\omega_i - \pi$   □.

According to Lemma 1, if $\varphi_1 - \varphi_2 > 0$, the system is stable and the range of $\omega_c$ can also be solved at the same time.

## 4. Verification of Dual-Locus Diagram Method

### 4.1. For First-Order Time-Delay Systems

The coupled tank is a typical industrial plant. Ravi Patel, Anil Gojiya, and Dipankar Deb used the DC gain technique to formulate failure reconfiguration of pumps in two reservoirs connected to the overhead tank [24]. To verify the correctness of the dual-locus diagram method, we conducted an experiment on coupled tanks. The level of the lower tank is controlled by changing the inflow rate of the upper tank, where the rate is controlled by a frequency converter. The plant can be regarded as a first-order plus time-delay process, where its gain, time constant, and delay time can be identified through the method of step response. The plant is controlled by PLC S7-300; thus, the LADRC algorithm must be discretized before being downloaded, where the discretization method of zero-order holder is used [25].

#### 4.1.1. Construction of Extended State Observer for First-Order Systems

We consider the following first-order system:

$$\dot{y} = g(t, y, u) + w + bu \tag{27}$$

where $u$ is the input, $y$ is the output, $b$ is relative gain, and $g$ and $w$ are internal disturbance and external disturbance, respectively.

We define $x_1 = y$ and $x_2 = f$, where $f$ is the total disturbance and $f = \dot{y} - b_0 u$. $b_0$ is the estimated value of $b$, and we can obtain the extended state space as follows:

$$\begin{aligned} \dot{x} &= Ax + bBu + Ef \\ y &= Cx + Du \end{aligned} \tag{28}$$

where $A = \begin{bmatrix} 0 & 1 \\ 0 & 0 \end{bmatrix}, B = \begin{bmatrix} 1 \\ 0 \end{bmatrix}, E = \begin{bmatrix} 0 \\ 1 \end{bmatrix}, C = \begin{bmatrix} 1 & 0 \end{bmatrix}, D = 0, x = \begin{bmatrix} x_1 \\ x_2 \end{bmatrix}$.

Constructing the state observer according to the given state space, we obtain

$$\begin{aligned} \dot{z} &= Az + b_0 Bu + L(y - \hat{y}) \\ \hat{y} &= Cz \end{aligned} \tag{29}$$

where $z = \begin{bmatrix} z_1 & z_2 \end{bmatrix}^T$ is the estimate of $x$.

Then, by assigning observer poles to observer bandwidth $\omega_o$:

$$\left| sI - (A - LC) \right| = (s + \omega_o)^2 \tag{30}$$

we can obtain

$$L = \begin{bmatrix} 2\omega_o \\ \omega_o^2 \end{bmatrix} \tag{31}$$

#### 4.1.2. Discretization of Extended State Observer

First, the continuous state space is discretized:

$$\begin{aligned} x(k+1) &= \Phi x(k) + \Gamma u(k) \\ y(k) &= Hx(k) + Ju(k) \end{aligned} \tag{32}$$

where $\Phi = e^{AT} = \begin{bmatrix} 1 & T \\ 0 & 1 \end{bmatrix}, \Gamma = \int_0^T e^{At} dt Bb_0 = b_0 \begin{bmatrix} T \\ 0 \end{bmatrix}, H = C, J = D = 0, T$ is the sampling period, which is equal to the scanning period of PLC.

An observer is constructed as follows:

$$\hat{x}(k+1) = \Phi \hat{x}(k) + \Gamma u(k) + L_d(y(k) - \hat{y}(k))$$
$$\hat{y}(k) = H\hat{x}(k) + Ju(k) \tag{33}$$

Replacing $L_d$ with $\Phi L_c$, we can obtain

$$\hat{x}(k+1) = \Phi \bar{x}(k) + L_d(y(k) - \hat{y}(k))$$
$$\bar{x}(k) = \hat{x}(k) + L_c(y(k) - \hat{y}(k)) \tag{34}$$

and then assigning observer poles to $\beta$:

$$\left| zI - (\Phi - L_dH) \right| = \left| zI - (\Phi - \Phi L_cH) \right| = (z - \beta)^2 \tag{35}$$

From the above equation, we know that $L_d = \Phi L_c = \begin{bmatrix} 2(1-\beta) \\ \frac{(1-\beta)^2}{T} \end{bmatrix}$ and $L_c = \Phi^{-1} L_d = \begin{bmatrix} 1 - \beta^2 \\ \frac{(1-\beta)^2}{T} \end{bmatrix}$. $\beta$ is the pole in the continuous system; thus, the pole in the discrete system is $\beta = e^{-\omega_o T}$ and $L_c = \begin{bmatrix} 1 - e^{-2\omega_o T} \\ \frac{(1-e^{-\omega_o T})^2}{T} \end{bmatrix}$ accordingly.

Second, the control law can be designed as follows:

$$u_0 = k_p(r - z_1) \tag{36}$$

$$u = \frac{u_0 - z_2}{b_0} \tag{37}$$

When the appropriate parameters are selected, the estimated value $z$ is almost equal to the actual value $x$, and $\dot{y}$ is close to $u_0$ accordingly.

According to the method of bandwidth parameterization, $k_p$ can be parameterized by $\omega_c$.

Third, the problem of how to implement the algorithms on PLC must be solved. We write the main discrete algorithm in "cyclic interrupt" organization block (OB35). At the beginning of the program, the calculation result of the previous scan cycle (x (k)) is moved to the current x (k + 1). When the current scan cycle ends, the new x (k + 1) is held until the next scan cycle. In this manner, the difference calculus can be implemented. The function block of the current discrete extended state observer (CDESO) is written first, and then the function block of LADRC is written on the basis of the CDESO.

In the experimental process, WinCC was used as the monitoring software. We selected CPU 315-2 PN/DP [6ES7 315-2EG10-0AB0] and SM334 AI/AO × 8/8Bit [SM334-0CE01-0AA0]. For the communication protocol between PLC and PC, we selected TCP/IP. The experimental schematic diagram and water tanks are as shown in Figures 3 and 4, respectively.

### 4.1.3. Analysis of Experiment Results

We first gave a "10 cm" signal manually, and then we changed the setting value to 15 cm. After many experiments, we obtained the following results: the delay time $\tau$ was 16 s, the system gain was 1.86, and time constant $T$ was 74 s.

Thus, the transfer function of the plant is $\frac{1.86}{74s+1} e^{-16s}$.

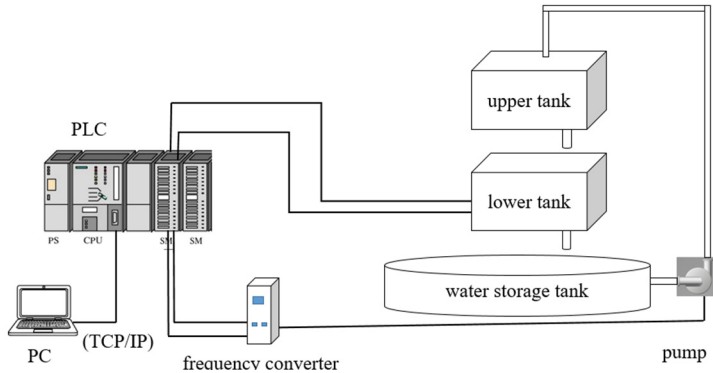

**Figure 3.** Experimental Schematic Diagram.

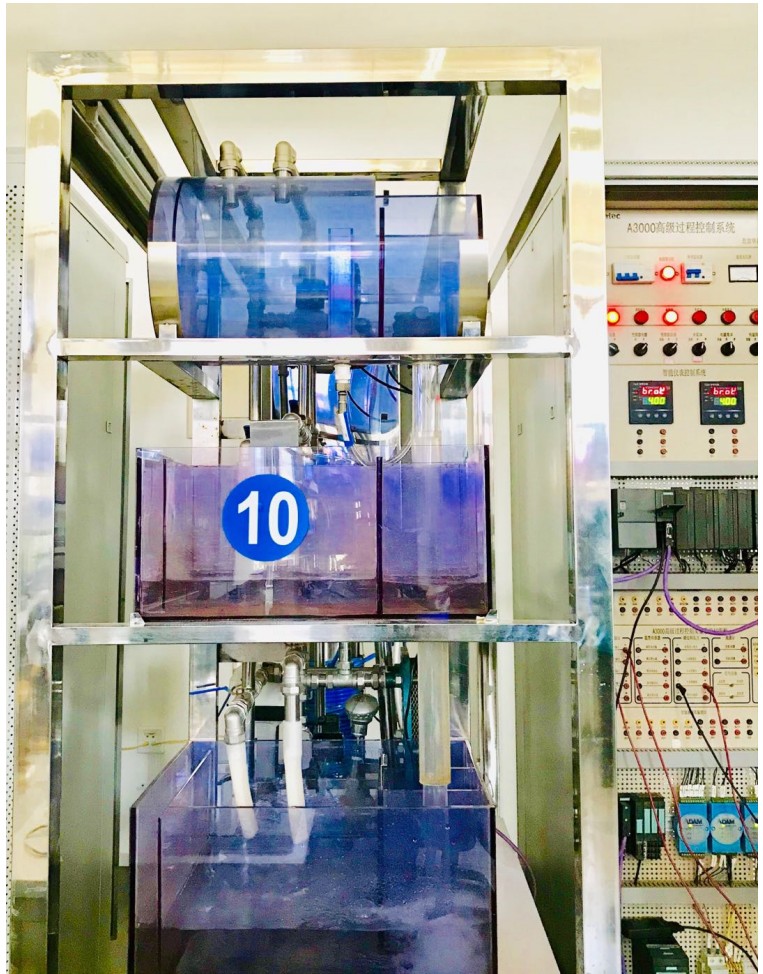

**Figure 4.** Water Tanks.

According to the dual-locus diagram method, the equation of frequency intersection $\omega_i$ is as follows in the LADRC controller corresponding to the first-order time-delay system:

$$\left|L(j\omega_i)\right| = \left|\frac{-\omega_c\omega_i{}^2 + \left(2K\omega_c^2 + K^2\omega_c^2\right)j\omega_i + K^2\omega_c{}^3}{\left(-\omega_i{}^2 + 2Kj\omega_c\omega_i\right)\left(j\omega_i T + 1\right)}\right| = 1 \tag{38}$$

The preceding equation can be simplified as

$$a\omega_i^6 + b\omega_i^4 + c\omega_i^2 - 1 = 0 \tag{39}$$

where $a = \frac{T^2}{K^4\omega_c{}^6}$, $b = \frac{\left(4K^2T^2\omega_c{}^2+1-\omega_c{}^2\right)}{K^4\omega_c{}^6}$, and $c = \frac{4K^2\omega_c^2+2K^2\omega_c^4-\left(2K\omega_c^2+K^2\omega_c^2\right)^2}{K^4\omega_c{}^6}$. [19] also presents its solutions that satisfy actual conditions:

$$\omega_i = \sqrt{\sqrt[3]{-\frac{q}{2} + \sqrt{\left(\frac{q}{2}\right)^2 + \left(\frac{p}{3}\right)^3}} + \sqrt[3]{-\frac{q}{2} - \sqrt{\left(\frac{q}{2}\right)^2 + \left(\frac{p}{3}\right)^3}} - \frac{b}{3a}} \tag{40}$$

where $p = \frac{c}{a} - \frac{b^2}{3a^2}$ and $q = \frac{2b^3}{27a^3} - \frac{bc}{3a^2} - \frac{1}{a}$.

The phase angle of $L(j\omega)$ at intersection frequency $\omega_i$ can be written as follows:

$$\varphi_1 = \arctan\left(\frac{2K\omega_c^2\omega_i + K^2\omega_c^2\omega_i}{K^2\omega_c{}^3 - \omega_c\omega_i^2}\right) - \arctan\left(\frac{\omega_i}{2K\omega_c}\right) - \arctan(T\omega_i) - \frac{\pi}{2} \tag{41}$$

Similarly, the phase angle of $-e^{j\omega\tau}$ at intersection frequency $\omega_i$ is $\varphi_2 = \tau\omega_i - \pi$. Let $\varphi_1 > \varphi_2$, and then the upper limit of $\omega_c$ can be obtained. We used MATLAB to perform this work. In these coupled-tank devices, $\omega_c$ is approximately 0.1505 when $K = 10$. First, we manually set the liquid level at 10 cm, where the set value of LADRC was also 10 cm. Then, we cut the manual mode of the controller to the automatic mode and changed the set value to 15 cm. We recorded the data and drew the figure by MATLAB (9.4, MathWorks, Natick, MA, US, 2018) and the result is shown in Figure 5.

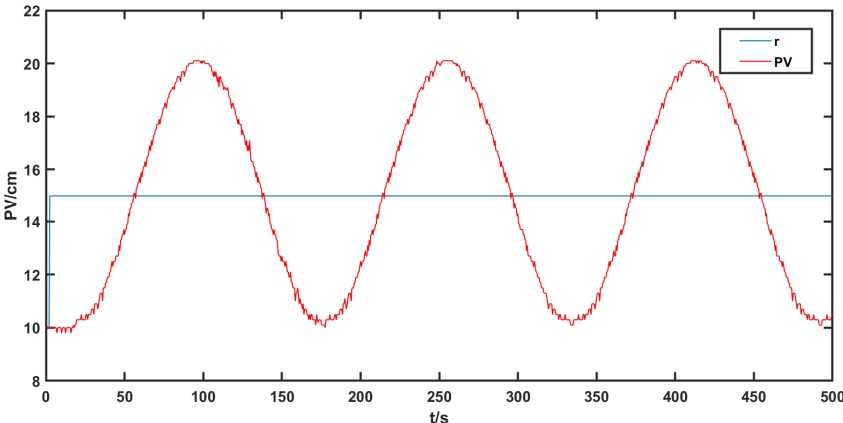

**Figure 5.** Step Response When $\omega_c = 0.1505$ and $K = 10$.

We can observe that the measured value PV of the level has an unattenuated oscillation between 10 cm and 20 cm. Also, the measured value of the level can be convergent if the controller bandwidth $\omega_c$ becomes smaller, which is verified in the next two experiments. This time we only changed $\omega_c$ to 0.1 and 0.05. The rest of the operation was the same as in the previous experiment. We obtained the following results:

According to Figures 6 and 7, a smaller controller bandwidth $\omega_c$ can make the plant convergent and lead to a smaller overshoot.

### 4.2. For Second-Order Time-Delay Systems

We considered a second-order time-delay system as follows:

$$G_p = G_0e^{-153s} = \frac{1.5}{(120s + 1)(75s + 1)}e^{-153s} \tag{42}$$

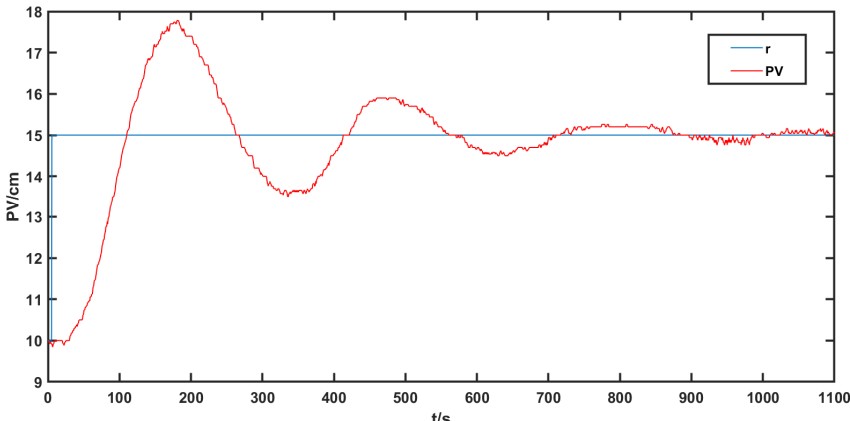

**Figure 6.** Step Response When $\omega_c = 0.1$ and $K = 10$.

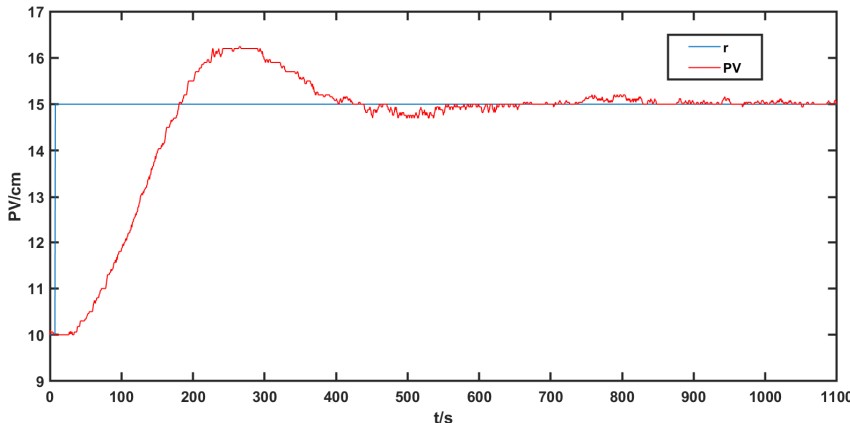

**Figure 7.** Step Response When $\omega_c = 0.05$ and $K = 10$.

The steady-state gain $b$ was 1.5, and two time constants $T_1$ and $T_2$ were 120 and 75, respectively. The delay time $\tau$ was 153 s. According to the theory introduced in part 2, a second-order LADRC controller can be chosen, and we selected $b_0 = b = 1.5$ and $K = 10$. In this way, Equation (22) becomes the following form:

$$L(s) = \frac{\omega_c^2 s^3 + \left(15\omega_c^3 + 150\omega_c^3 + 125\omega_c^3\right)s^2 + \left(75\omega_c^4 + 250\omega_c^4\right)s + 125\omega_c^5}{s\left(s^2 + 15\omega_c s + 2\omega_c s + 75\omega_c^2 + 30\omega_c^2\right)(120s + 1)(75s + 1)} = -e^{153s} \tag{43}$$

After simplification, the preceding equation becomes

$$L(s) = \frac{\omega_c^2 s^3 + 290\omega_c^3 s^2 + 325\omega_c^4 s + 125\omega_c^5}{s\left(s^2 + 17\omega_c s + 105\omega_c^2\right)(120s + 1)(75s + 1)} = -e^{153s} \tag{44}$$

Let its modulus be equal to the modulus of $-e^{153s}$, and we can obtain the following equation and solve for $\omega_i$, which is a function of $\omega_c$:

$$\left|L(j\omega_i)\right| = \left|\frac{-j\omega_c^2\omega_i^3 - 290\omega_c^3\omega_i^2 + j\omega_c^4\omega_i + 125\omega_c^5}{j\omega_i\left(-\omega_i^2 + 17j\omega_c\omega_i + 105\omega_c^2\right)(120j\omega_i + 1)(75j\omega_i + 1)}\right| = \left|-e^{153j\omega_i}\right| = 1 \tag{45}$$

From the introduction in part 2, the phase angles of $L(j\omega)$ and $-e^{153j\omega}$ can be written as

$$\varphi_1 = \arctan\left(\frac{-\omega_c^2\omega_i^3 + \omega_c^4\omega_i}{-290\omega_c^3\omega_i^2 + 125\omega_c^5}\right) - \frac{\pi}{2} - \arctan\left(\frac{17\omega_c\omega_i}{-\omega_i^2 + 105\omega_c^2}\right) - \arctan(75\omega_i) - \arctan(120\omega_i) \quad (46)$$

$$\varphi_2 = \tau\omega_i - \pi \quad (47)$$

Furthermore, their difference value is

$$\varphi_1 - \varphi_2 = \arctan\left(\frac{-\omega_c^2\omega_i^3 + \omega_c^4\omega_i}{-290\omega_c^3\omega_i^2 + 125\omega_c^5}\right) - \arctan\left(\frac{17\omega_c\omega_i}{-\omega_i^2 + 105\omega_c^2}\right)$$
$$-\arctan(75\omega_i) - \arctan(120\omega_i) - 153\omega_i + \frac{\pi}{2}$$

where $\omega_i$ is a function of $\omega_c$; thus, $\varphi_1 - \varphi_2$ is also a function of $\omega_c$.

We can find that solving for $\omega_c$ is equivalent to solving the following nonlinear system of equations:

$$\begin{cases} |L(j\omega_i)| = \left|\dfrac{-j\omega_c^2\omega_i^3 - 290\omega_c^3\omega_i^2 + j\omega_c^4\omega_i + 125\omega_c^5}{j\omega_i(-\omega_i^2 + 17j\omega_c\omega_i + 105\omega_c^2)(120j\omega_i + 1)(75j\omega_i + 1)}\right| = \left|-e^{153j\omega_i}\right| = 1 \\ \varphi_1 - \varphi_2 = \arctan\left(\dfrac{-\omega_c^2\omega_i^3 + \omega_c^4\omega_i}{-290\omega_c^3\omega_i^2 + 125\omega_c^5}\right) - \arctan\left(\dfrac{17\omega_c\omega_i}{-\omega_i^2 + 105\omega_c^2}\right) - \arctan(75\omega_i) \\ \qquad\qquad -\arctan(120\omega_i) - 153\omega_i + \dfrac{\pi}{2} = 0 \end{cases} \quad (48)$$

where the two unknown variables $\omega_c$ are and $\omega_i$. In fact, the preceding nonlinear system of equations means that the argument and modules of $L(j\omega_i)$ are equal to the argument and modules of $-e^{153j\omega_i}$, respectively. In other words, the real and imaginary parts of $L(j\omega_i)$ are equal to the real and imaginary parts of $-e^{153j\omega_i}$, respectively. In addition, we can know that $-e^{153j\omega_i} = -\cos(153\omega_i) - j\sin(153\omega_i)$ from Euler's formula. Then, another nonlinear system of equations containing trigonometric functions, which is equal to Equation (48), can be obtained:

$$\begin{cases} 900\omega_c\omega_i^4 - 9135\omega_c^2\omega_i^2 + 87\omega^4 - 17\omega_c\omega_i^2 = (290\omega_c^3\omega_i^2 - 125\omega_c^5)\cos(153\omega_i) + (\omega_c^2\omega_i^3 - \omega_c^4\omega_i)\sin(153\omega_i) \\ -94500\omega_c^2\omega_i^3 + 900\omega^5 - 1479\omega_c\omega_i^3 + 105\omega_c^2\omega_i - \omega_i^3 = (125\omega_c^5 - 290\omega_c^3\omega_i^2)\sin(153\omega_i) + (\omega_c^2\omega_i^3 - \omega_c^4\omega_i)\cos(153\omega_i) \end{cases} \quad (49)$$

Systems of equations in the preceding forms can be solved by the function fsolve in MATLAB. Then, we can obtain $\omega_c \approx 0.1727$ under the condition of $K = 5$, and its step response is shown in Figure 8.

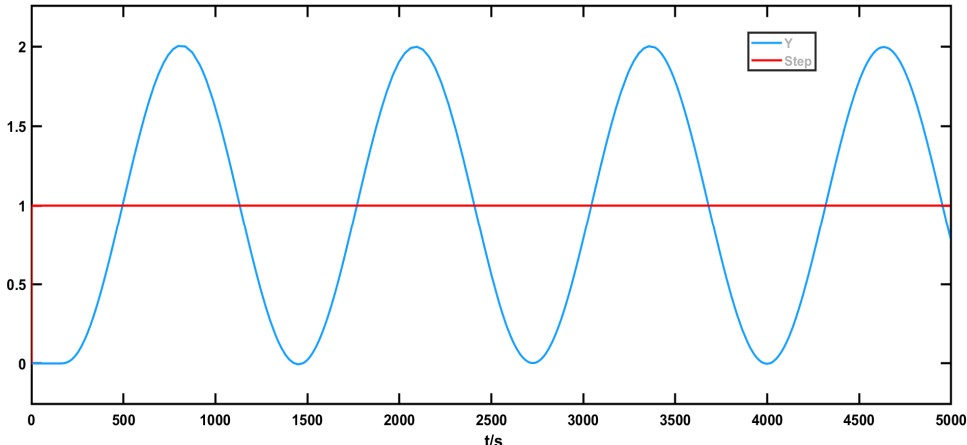

**Figure 8.** Step Response of Second-Order Time-Delay System When $\omega_c \approx 0.1727$ and $K = 5$.

At the same time, we can obtain the Nyquist curves of $L(s)$ and $-e^{153s}$ as shown in Figure 9.

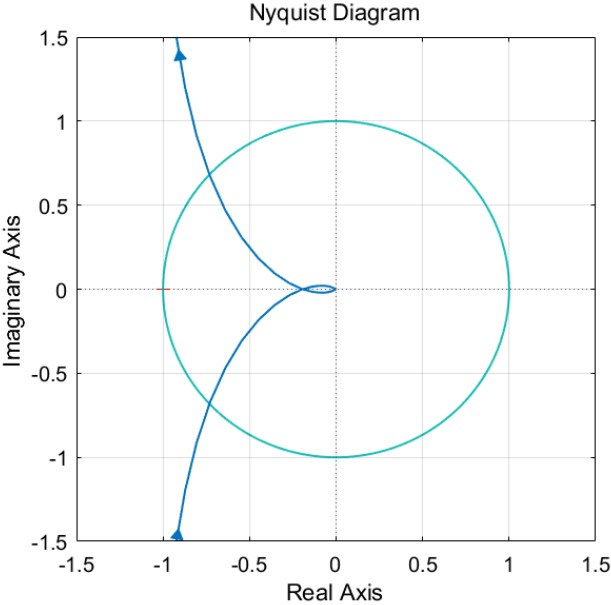

**Figure 9.** Dual-Locus Diagram of $L(s)$ and $-e^{153s}$.

When $K$ takes different values, $\omega_c$ will have different stability regions. Using the above method, we calculate the stability region when $K$ takes different values. The results are shown in Figure 10.

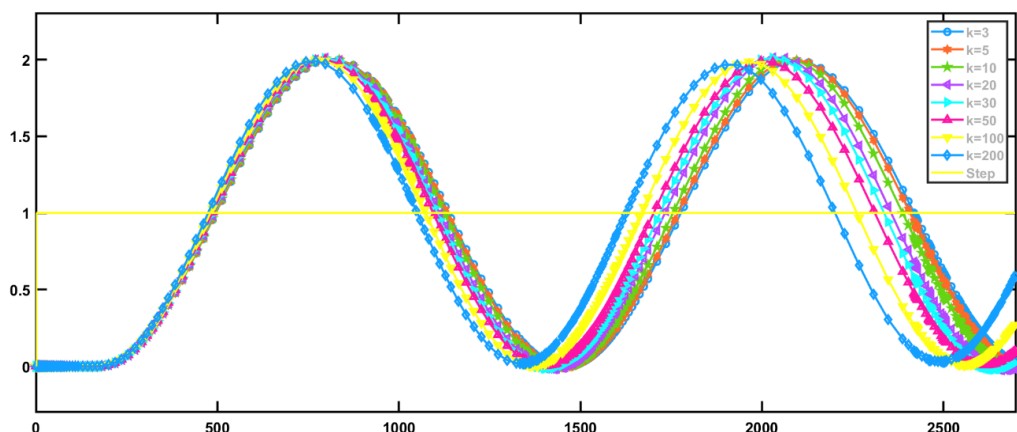

**Figure 10.** Step Response (Equal Amplitude Oscillation) of a Second-Order Time-Delay System for Different $K$.

The local enlargement is shown in Figure 11.

The corresponding upper bounds of $\omega_c$ and $\omega_o$ are shown in Table 1 when $K$ takes different values in the preceding figures.

**Table 1.** Upper Limit of $\omega_c$ for Different $K$.

| $K$ | 3 | 5 | 10 | 20 | 30 | 50 | 100 | 200 |
|---|---|---|---|---|---|---|---|---|
| $\omega_c$ | 0.2164 | 0.1727 | 0.1308 | 0.1016 | 0.0882 | 0.0741 | 0.05905 | 0.0473 |
| $\omega_o$ | 0.6492 | 0.8635 | 1.308 | 2.032 | 2.646 | 3.705 | 5.905 | 9.46 |

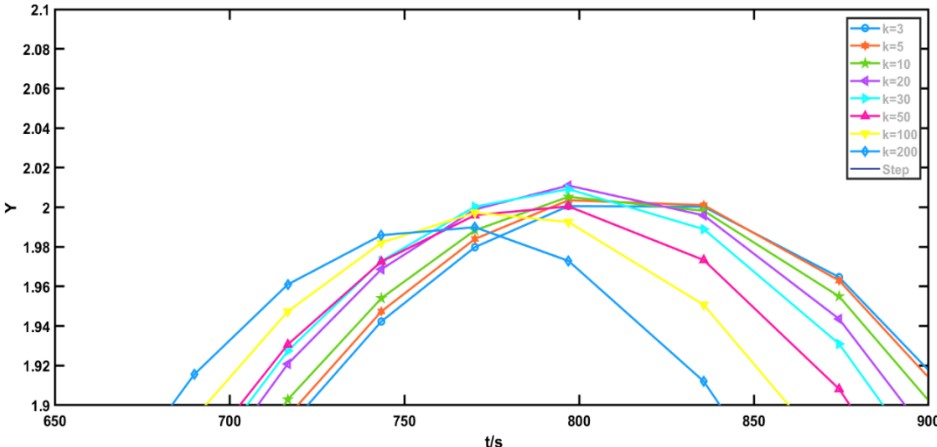

**Figure 11.** Local Magnification for Figure 10.

From Figure 11 and Table 1, the upper limit of $\omega_c$ decreases with the increase of *K*. However, its amplitude and period of step response decrease with the increase of *K*, which means that the increase of *K* helps improve the control performance.

In the actual industrial process, the devices cannot be permitted to work at the upper limit of $\omega_c$ because it will wear the devices seriously. In fact, the overshoot and setting time are expected to be as small as possible for the actual control performance. When *K* is given, we can reduce the overshoot by reducing $\omega_c$, and the relationship between system step response and $\omega_c$ is shown in Figure 12.

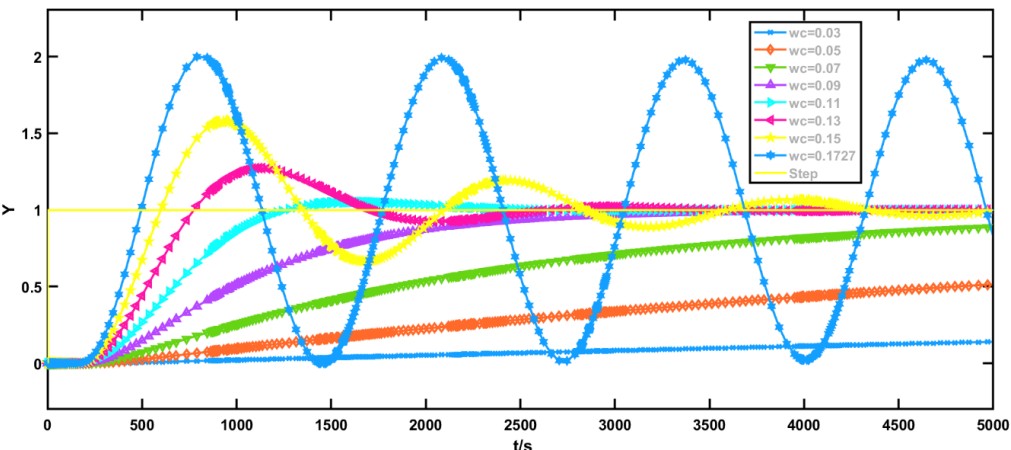

**Figure 12.** Step Response of a Second-Order Time-Delay System for Different $\omega_c$ and *K* = 5.

As shown in Figure 12, the overshoot decreases with the decrease of $\omega_c$. However, the setting time also lengthens with the decrease of $\omega_c$. Thus, considering all these factors, we find that $\omega_c$ is acceptable between 0.09 and 0.11 in the actual process.

When $\omega_c$ is in the aforementioned range, its step response has a relatively shorter setting time and smaller overshoot. To determine the optimal value of $\omega_c$, GA is proposed [26]. As the upper limit of $\omega_c$ can be solved by the method of dual-locus diagram, $\omega_c$ can quickly converge to the optimal value, which is a problem about multiple local optima in the objective space.

A suggested fitness function is as follows:

$$J = \int_0^\infty \left( \omega_1 t |e(t)| + \omega_2 u^2(t) \right) dt + \omega_3 u_{max} + \omega_4 \delta_p \tag{50}$$

where $e$, $u$, $u_{\max}$, and $\delta_p$ are error, control quantity, maximum value of control quantity, and overshoot of the system, respectively. Weights $\omega_1$, $\omega_2$, $\omega_3$, and $\omega_4$ are 1, 1, 5,000,000, and 5,000,000, respectively. The range of parameter $\omega_c$ is 0–0.1727 when $K = 5$, and the population size is 50. The stopping generation is 50. The results are presented in Figure 13.

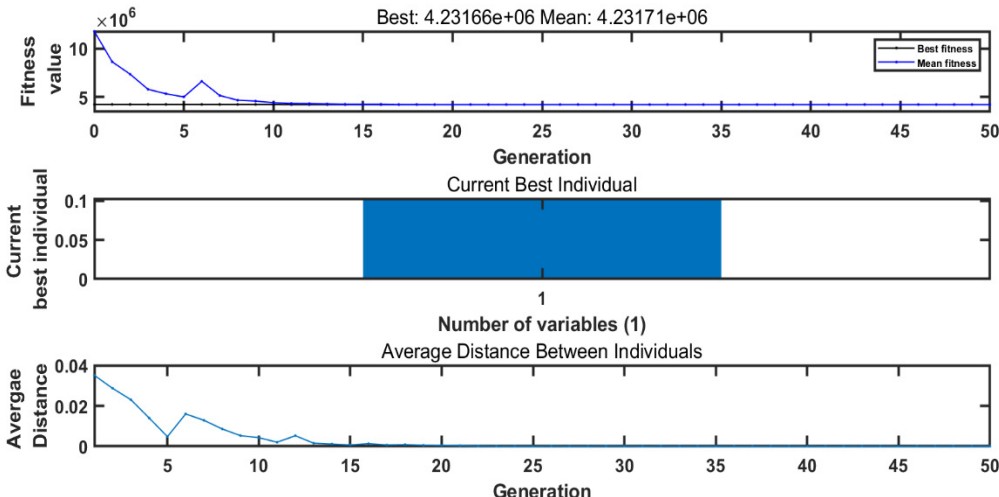

**Figure 13.** Process of Objective Function Optimization for $\omega_c$.

Figure 13 shows that $\omega_c$ converges to the optimal value at the 15th generation, and the best individual is 0.103. The step response for $\omega_c = 0.103$ is shown in Figure 14.

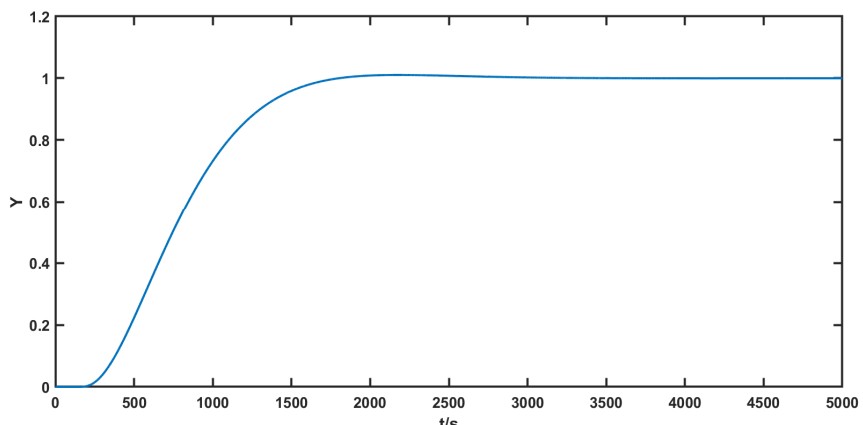

**Figure 14.** Step Response of a Second-Order Time-Delay System for $\omega_c = 0.103$ when $K = 5$.

Figure 14 shows that the step response of the system has a relatively shorter setting time and smaller overshoot when $\omega_c = 0.103$.

In the preceding section, we discussed the effect of the change of $\omega_c$ in control performance when $K$ is given. Also, we examined the effect of the change of $K$ on the control performance when $\omega_c$ is given. The next simulation was used to discuss the above issues. In this simulation, $\omega_c$ was set to 0.0473, which is the upper limit for $K = 200$. We selected $K$ as 3, 5, 10, 20, 30, 50, 100, and 200 to perform simulations for its step response. The results are shown in Figure 15.

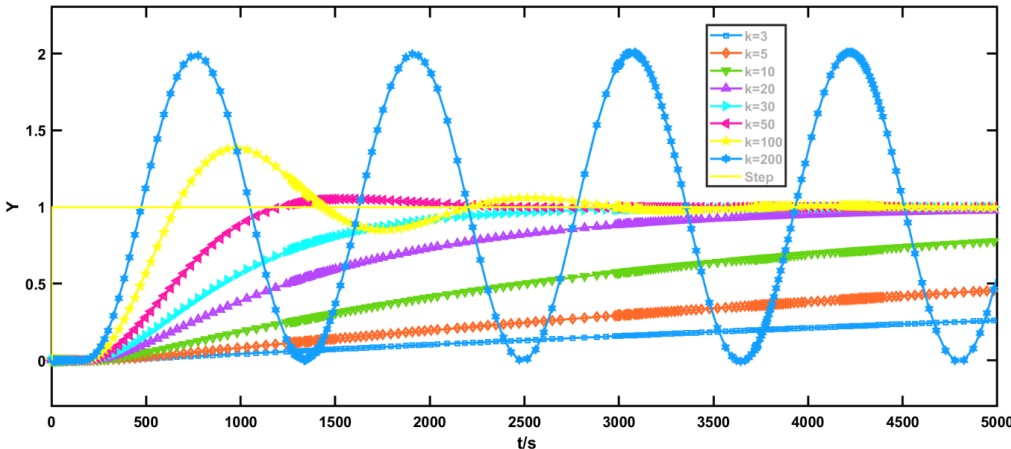

**Figure 15.** Step Response of a Second-Order Time-Delay System for Different *K* when $\omega_c$ = 0.0473.

Figure 15 shows that the overshoot decreases with the decrease of *K*, but similar to $\omega_c$, a smaller *K* means a longer setting time. Furthermore, the acceptable range of *K* is 50–100. Evidently, the effect of the change of *K* on the control performance when $\omega_c$ is given is similar to $\omega_c$. However, *K* is not supposed to be extremely large in the actual process. Similarly, the optimal value of *K* can be found by GA, which is also a problem about multiple local optima in the objective space. The process of objective function optimization for *K* is shown in Figure 16.

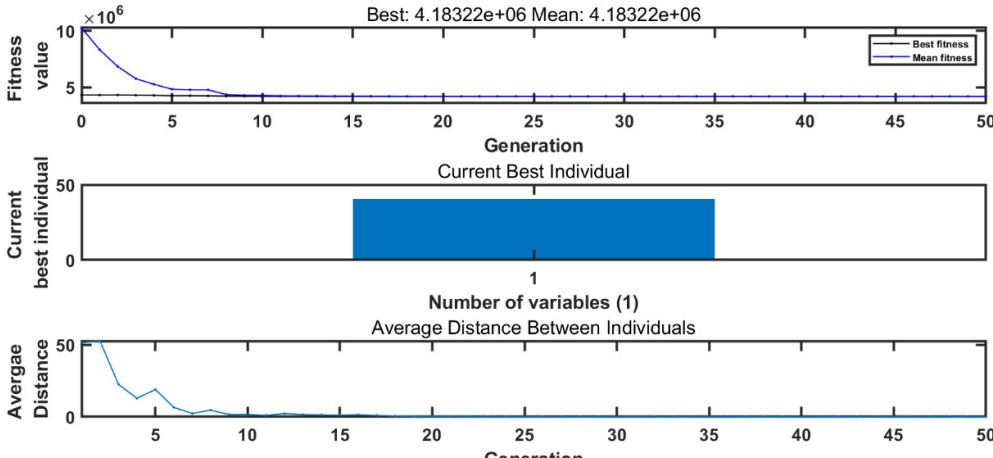

**Figure 16.** Process of Objective Function Optimization for *K*.

Figure 16 shows that *K* converges to the optimal value at the 10th generation, and the best individual is 40.626. The step response for *K* = 40.626 is presented in Figure 17.

Figure 17 shows that the step response of the system has a relatively shorter setting time and smaller overshoot when *K* = 40.626.

From the preceding analysis, we can observe that an ideal control performance needs an appropriate controller bandwidth $\omega_c$ and bandwidth ratio *K*. For the actual process, we can first determine the approximate range of *K*, and then select the appropriate $\omega_c$. When we select roughly $\omega_c$ and *K*, we can fine-tune them to find the best values according to the actual process.

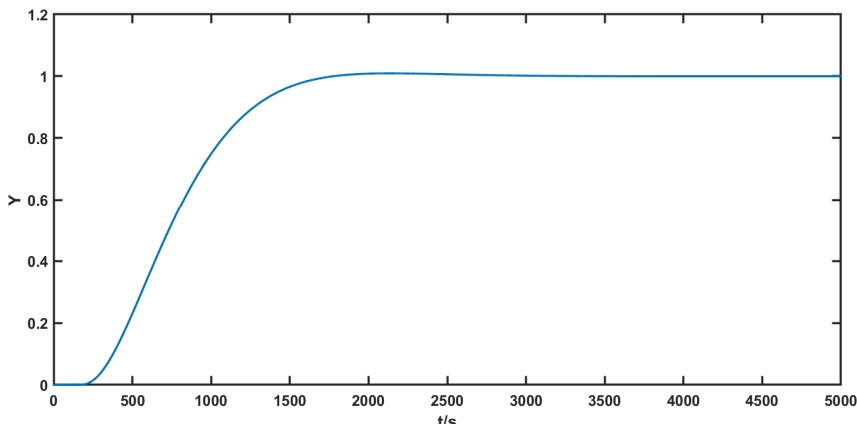

**Figure 17.** Step Response of Second-Order Time-Delay System for $\omega_c = 0.0473$ when $K = 40.626$.

## 5. Conclusions

In this study, the dual-locus diagram method for the parameter stability region of first-order time-delay system LADRC controllers is validated by water tank experiments by PLC. A method for solving the parameter stability region of LADRC controllers of second-order time-delay systems is also provided, and simulations are conducted to validate this method. GA is introduced to optimize the LADRC controller parameters and achieves good results. These experimental and simulation results show that the actual process needs an appropriate bandwidth ratio $K$ and controller $\omega_c$, and smaller $\omega_c$ and $K$ both mean a smaller overshoot and longer setting time. The adjustable range of $K$ is relatively smaller in the actual process and the adjustable range of $\omega_c$ has an upper limit. In the actual process, both $K$ and $\omega_c$ need to be considered. However, the method of solving the upper limit of $\omega_c$ complicates the application of LADRC on second-order time-delay systems. Although GA can find the optimal parameter, further effort is needed to simplify the dual-locus diagram method for its application.

**Author Contributions:** D.L. and X.C. conceived and designed the experiments; X.C. and J.Z. performed the experiments; X.C. analyzed the data; J.Z. and Q.J. contributed devices and analysis tools; D.L. and X.C. wrote and edited the paper. All authors have read and agreed to the published version of the manuscript.

**Funding:** This work was supported in part by the National Natural Science Foundation of China under Grant 61873022, 61573052, and Beijing Natural Science Foundation under Grant 4182045.

**Conflicts of Interest:** This research received no external funding.

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
