# Peer review of "On Parameter Stability Region of LADRC for Time-Delay Analysis with a Coupled Tank Application"

_processes, doi:10.3390/pr8020223_

Round 1

Reviewer 1 Report

## General Comments

I hope it is not an issue of the journal but I found the manuscript barely
readable because of its bad formatting. (see the screenshot below for a
particular bad exerpt). Also, it would be good if intertext links were working
in the pdf. The language might need some editing but that's not a major issue.
More important would be a better formulation of the content, notably Theorem 1
should be formulated like a mathematical theorem.

Having said this, I think that the content of the manuscript might be suitable
for publication since the problem class is relevant and since applicability has
been proven in an experiment.

## Particular Issues

### Chapter 3

This is the major theoretical content. However it is very unclear what results
are valid for a general mathematical model and what results are only good for
the particularly chosen plants. This should be better specified.

### In Chapter 3.1

* there should be a reference for Lemma 1

### In Chapter 3.2

* Write down the consider system in time domain
* There is no $G_p$ in Figure 2
* the `*` has not been introduced -- is it the convolution operator?
* somewhat later, the `x` (multiplication operator?) has not been introduced,
either
* I don't see a delay in (13), please explain
* Theorem 1 needs to be stated more *mathematically*

### References

* except from a textbook, there are only references from authors that,
obviously, are connected to the Chinese education and research environment.
This is not a problem a priori, however a good overview of the
state-of-the-art certainly would include achievements of other research
communities.

Reviewer 2 Report

This manuscript appears to extend the authors' previous work on the stability analysis of LADRC for first order systems with time delay. Please note that the reviewer is not extremely familiar with industrial process control. Nevertheless, there are a number of points that might reasonably be clarified:

1. The authors may wish to define SISO (assumed "single input single output") before the first use of the term.

2. In Equations 1, 3, 4 & 5, the authors may wish to more explicitly define what a dot above certain variables indicates (this may be a known convention in this subfield, but may not be immediately clear to others)

3. In Equation 7, the variables k and r do not seem to be defined. Moreover, more details may be added as to the choice of this formulation. Equation 12 later seems to suggest that k is also an undetermined coefficient; if so, it might be stated. Also, later in Theorem 1, k is defined as the appropriate bandwidth ratio. This might be renamed to avoid possible confusion with the same symbol corresponding to different meanings.

4. In Line 128, the statement "we can convert L_2(s) to e^rs" might be further explained, especially since e^rs seems to be used as the second-order time delay formulation in Section 4.2.

5. For Line 169, more details could be provided for the "theory of bandwidth parameterization", or a reference cited, especially since it is also referenced in later proofs/equations.

6. In Line 216, it is stated that "From Eq. (22), we know that the order of L(s) is 5"; should it be Equation 23 instead?

7. For Section 4.2, the second-order time delay system seems to correspond to an actual industrial process (Line 376), possibly something like the water tank example given for first-order time delay in Section 4.1. Was the second-order system function (Equation 42) obtained/approximated from that actual process, and if so, how was it approximated?

8. Still for Section 4.2, is there any reason bandwidth ratio k=5 was selected before searching the controller bandwidth w_c parameter space (Figure 14), and w_c=0.0473 was selected before selecting the k space, instead of a grid search of k & w_c together? In particular, the "optimal combination of parameters" seems to be different depending on whether k or w_c is explored first; when setting k=5 and w_c is explored, the optimal combination is k=5, w_c=0.103 (Line 398), and when setting w_c=0.0473 and k is explored, the optimal combination is k=40.626, w_c=0.0473. If this represents multiple local optima in the objective space (instead of a global optimum), the authors might explicitly state so.

9. In Line 419, should it be "The step response for k=40.626" instead of "w_c=40.626"?

10. For Reference 8, there seems to be an error in the page number ("574-580" instead of "674-580")

Reviewer 3 Report

Inserting formulas into the text resulted in slippage.

No space left after [12] references.

The caption in Figure 4 is different from the rest.

The tests were performed with specific parameters.
It would also be useful to formulate general relationships.

Round 2

Reviewer 1 Report

I'm fine with all changes.

Author Response

Answer Sheets

Manuscript Number: processes-700197

Paper Title: Research on the Parameter Stability Region of LADRC for Time-Delay Systems

Authors: Dazi Li, Xun Chen, Jiangqing Zhang and Qibing Jin

We would like to express our sincere thanks to the reviewer for the constructive and positive comments. Thank you for your affirmation of the research work of this paper. We earnestly appreciate for the editor and reviewer’s help.

Thank you again for your help.

Authors

2020/2/3

Reviewer 2 Report

We thank the authors for considering our previous comments. There remains the minor issue of the first use of SISO being around Line 54, instead of line 84 where the full text of the abbreviation is currently defined.

Author Response

(The authors gave the same response as above.)
